# RELATIONAL MULTI-INSTANCE LEARNING FOR CONCEPT ANNOTATION FROM MEDICAL TIME SERIES

## ABSTRACT

Recent advances in computing technology and sensor design have made it easier to collect longitudinal or time series data from patients, resulting in a gigantic amount of available medical data. Most of the medical time series lack annotations or even when the annotations are available they could be subjective and prone to human errors. Earlier works have developed natural language processing techniques to extract concept annotations and/or clinical narratives from doctor notes. However, these approaches are slow and do not use the accompanying medical time series data. To address this issue, we introduce the problem of concept annotation for the medical time series data, i.e., the task of predicting and localizing medical concepts by using the time series data as input. We propose Relational Multi-Instance Learning (RMIL) - a deep Multi Instance Learning framework based on recurrent neural networks, which uses pooling functions and attention mechanisms for the concept annotation tasks. Empirical results on medical datasets show that our proposed models outperform various multi-instance learning models.

## 1 INTRODUCTION

Clinicians have limited time (e.g., only a few minutes (Howie et al., 1999)) to study and treat each patient. However, they are overloaded with a lot of patient data from multiple sources and in various formats, such as patient medical history and doctor's notes in free-flowing text, vitals and monitoring data which are captured as time series, and prescriptions and drugs which appear as medical codes including ICD-9 (Organization & Corporation, 1998), LOINC codes (Forrey et al., 1996), etc. This rich information should be summarized and available to clinicians in easily digestible format for faster diagnosis and treatment. Graphical visualizations (Plaisant et al., 1998) are a popular approach to show patient data to doctors. However, recent studies have shown that graphical visualisations are not always helpful for clinicians' decision-making (Law et al., 2005; Van Der Meulen et al., 2010). Text summaries on the other hand are widely embraced and are usually adopted in practice (Scott et al., 2013). Most existing systems use natural language processing techniques (Afantenos et al., 2005; Giordano et al., 2015) to generate summaries from doctor notes which include test results, discharge reports, observational notes, etc. While these systems are useful, they only use one source of data, i.e., doctor's notes which might have noisy and erroneous entries, for text summarization. On the other hand, electronic health records have other sources of patient data such as vital signs, monitoring sensors, and lab results in the form of multivariate time-series, which can be more accurate and may contain rich information about patient's conditions. Few existing patient summarization systems actually extract information directly from these time series for concept prediction and/or summarization. Generating simple text summaries such as trends from time series has been investigated before (Sripada et al., 2003) but is marginally useful since these trends are not mapped to the medical concepts which clinicians can quickly comprehend. Recent works (Pham et al., 2016; Choi et al., 2016a;b; Lipton et al., 2015; Che et al., 2016) have successfully shown that clinical events and outcomes can be predicted using medical codes or clinical time series data. However, directly obtaining medical concept annotations and summaries from the time series data is still an open question.

In this work, we introduce the concept annotation task as the problem of predicting and localizing the medical concepts by modeling the related medical time series data. Figure 1 illustrates a concept annotation example where medical time series data such as heart rate, pH and blood gas pressure are given, and the goal is to predict the time series of concepts such as intubation, extubation and

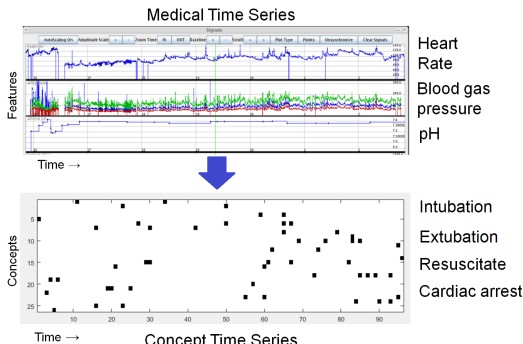

**Figure 1:** Medical time series to concept annotations.

resuscitate. To solve concept annotation problem, we formulate it as a Multi-Instance Learning (MIL) problem (Dietterich et al., 1997) and propose a deep learning based framework called Relational Multi-Instance Learning (RMIL). RMIL uses Recurrent Neural Networks (RNNs) to model multivariate time series data, leverages instance relations via attention mechanisms, and provides concept predictions using pooling functions.

The main contributions of our work are the following. We present a unified view of the MIL approaches for time series data using RNN models with different pooling functions and attention mechanisms. We show that our RMIL model is capable of learning a good classifier for concept detection (bag label predictions) and concept localization tasks (instance label prediction), even though it is only trained using bag labels. We demonstrate that RMIL obtains promising results on real-world medical datasets and outperforms popular MIL approaches.

The rest of the paper is structured as follows. In the following section, we briefly discuss the related works. Afterwards, we describe MIL framework and describe how RNN can be combined with multi-instance learning framework to obtain our proposed RMIL. In Sections 4 and 5, we present experimental results and conclusions respectively. In the appendix, we demonstrate anomaly detection as another application of our RMIL framework.

## 2 RELATED WORK

Discovering concept annotations from the multivariate time series is a relatively new problem in medical domain with limited prior work. In this section, we will first highlight the related works on annotation tasks and then review related works on multi-instance learning.

**Concept Annotation**    In medical domain, concept annotation is usually addressed in the clinical narrative mining and biomedical text mining literature (Aggarwal & Zhai, 2012; Cohen & Hersh, 2005; Zweigenbaum et al., 2007; Vincze et al., 2008). In other domains such as web-mining and computer vision, the concept annotation is usually analogous to semantic annotation (Kiryakov et al., 2004), image annotation (Jeon et al., 2003), object localization (Lampert et al., 2008) and image captioning (Karpathy et al., 2014).

*Clinical Narratives Mining* Automated discovery of temporal relations from clinical narratives (Savova et al., 2009; Zhou et al., 2006; Albright et al., 2013) and doctor notes (Plaisant et al., 1996) to uncover the patterns of disease progression is an important research problem in clinical informatics. Recent efforts such as SemEval competitions (Bethard et al., 2016) have been conducted to study this problem and evaluate/benchmark clinical information extraction systems (Xu et al., 2010). These competitions focus on discrete, well-defined tasks which allow for rapid, reliable and repeatable evaluations. However, they only consider identifying and extracting temporal relations from clinical notes and do not use the accompanying medical time series data.

*Image Annotation and Captioning* Successful object recognition systems have been developed in the past few decades for image annotation, object detection and localization in images and videos. ImageNet (Deng et al., 2009) and PASCAL challenges (Everingham et al., 2010) have greatly

accelerated the research in this area. Image captioning and visual-to-text translation, which are more generalized image annotation tasks, have been recently studied in several works (Karpathy et al., 2014; Kojima et al., 2002; Mao et al., 2014; Yu et al., 2016) where the goal is to find a text caption for a given image. Deep learning models such as RNN and sequence-to-sequence models (Sutskever et al., 2014) have achieved excellent results for image annotation/captioning tasks.

**Multi-Instance Learning**   Multi-Instance Learning (MIL), a well known researched topic in machine learning, was first introduced by Dietterich et al. (1997) as a form of weakly supervised learning for drug activity prediction. MIL frameworks have since been applied to many other domains including image and text annotations (Chen & Wang, 2004; Soleimani & Miller, 2017). Andrews et al. (2003) adapted Support Vector Machines (SVM) to the MIL framework and introduced miSVM and MISVM for optimizing instance-level and bag-level classifications respectively. Zhou & Zhang (2007) further extended MIL and proposed MIMLSVM for tackling multi-label problems. Zhou et al. (2009) introduced miGraph and MIGraph to model the structure in each bag. Zhang et al. (2011); Guan et al. (2016) also proposed MIL framework for structure data by leveraging the relational structures at the bag and instance levels. Generative model based MIL frameworks such as Multi-Instance Mixture Model (MIMM)(Foulds & Smyth, 2011), and Dirichlet Process Mixture of Guassians (DPMIL) (Kandemir & Hamprecht, 2014) have also been proposed for binary multi-instance classification. Guan et al. (2016) used an autoregressive hidden Markov model and proposed an MIL framework for activity recognition in time-series data. Garcez & Zaverucha (2012) used recurrent neural networks to combine instance-level preprocessing and bag-level classification in MIL setting. Comprehensive reviews of MIL approaches are provided in Amores (2013); Herrera et al. (2016); Soleimani & Miller (2017). Recently, deep learning models have been successfully applied for MIL framework (Zhu et al., 2017; Wu et al., 2015; Huang et al., 2013; Yan et al., 2016; Kotzias et al., 2014; Kraus et al., 2016) and these approaches are generally termed as deep multi-instance learning models. Most of these works use either convolutional neural networks or deep neural networks in their MIL framework for image annotation, labeling, segmentation or classification tasks. Despite the popularity of deep models for MIL, there are few works which have extended deep MIL models for multivariate time series data. The goal of this paper is to propose and study deep multi-instance learning models for multivariate time series data.

## 3   METHODS

In this section, we will first describe the Multi-instance learning framework, and then present our problem formulation and our proposed relational multi-instance learning models.

### 3.1   MULTI-INSTANCE LEARNING

Multi-instance learning (MIL) is a form of weakly supervised learning where the training data is arranged in sets called *bags*, and a label is provided for the entire bag. The data points inside a bag are referred to as *instances*. In the MIL framework, instance labels are not provided during training. The main goal of MIL is to learn a model based on the instances in the bag and the label of the bag - to make bag-level and instance-level predictions. In this work, we only focus on the classification task in MIL, leaving out other learning tasks such as regression. Generally, two broad assumptions can be used to model the relationship between instance label and bag label. In the standard MIL assumption (Dietterich et al., 1997), the bag label is negative if all the instances in the bag have a negative label, and the bag label is positive if at-least one of the instances in the bag has a positive label. Following the notations of Carbonneau et al. (2017), let $X$ denote a bag with $N$ feature vector instances i.e., $X = \{x_1, ..., x_N\}$. Let each instance $x_i$ in feature space $\mathcal{X}$ be mapped to a class by some process $f : \mathcal{X} \rightarrow \{0, 1\}$, where 0 and 1 correspond to negative and positive labels respectively. The bag classifier, also know as the aggregator function, $g(X)$ is defined by:

$$g(X) = \begin{cases} 1 & \text{if } \exists x \in X : f(X) = 1; \\ 0 & \text{otherwise} \end{cases}$$

The standard assumption is quite restrictive for some problem settings, where the positive bags cannot be identified by a single instance. Thus, this assumption can be relaxed to a collective assumption

which says that several positive instances in a bag are necessary to assign a positive label to that bag. In this case, a bag classifier is given by:

$$g(X) = \begin{cases} 1 & \text{if } \sum_{x \in X} f(X) \geq \theta; \\ 0 & \text{otherwise} \end{cases}$$

where $\theta$ is a threshold which indicates the minimum number of instances with positive labels that should be present in a bag to assign a positive label to that bag Weidmann et al. (2003). As discussed in section 2, a plethora of works have adapted machine learning models to the MIL setting to optimize instance-level and/or bag-level predictions.

### 3.2 PROBLEM FORMULATION

We formulate the concept annotation task as the detection and localization of concepts given the medical time series data. Let each patient $i \in \{1, .., N\}$ be associated with a medical time series (also referred to as feature time series) denoted by $\boldsymbol{X}_i \in \mathbb{R}^{T \times D}$, where $D$ denotes the number of features (such as heart rate, blood pressure) and $T$ denotes the length of time series observations (i.e., amount of time a patient is monitored). Let $\mathcal{C}$ denote the set of all the concepts associated with the $N$ patients, and $\boldsymbol{Y}_i \in \{0,1\}^K$ denote the concepts associated with $i^{th}$ patient where $K = |\mathcal{C}|$. Let $\boldsymbol{C}_i \in \{0,1\}^{T \times K}$ denote the concept time series of $\boldsymbol{X}_i$ with $C_i^{jk} = 1$ when concept $k$ is present at time-stamp $j$. In multi-instance learning settings, we treat each time series $\boldsymbol{X}_i$ as one bag, and the observation at each time step $j$ i.e. $\boldsymbol{X}_i^j \in \mathbb{R}^D$ as an instance in that bag. We are interested in the following tasks:

- Concept prediction task: for patient $i$, given $\boldsymbol{X}_i$, predict $\boldsymbol{Y}_i$.
- Concept localization task: for patient $i$, given $\boldsymbol{X}_i$, predict $\boldsymbol{C}_i$.

Notice that during training phase, only the input $\boldsymbol{X}_i$ and prediction label $\boldsymbol{Y}_i$ are available. Though $\boldsymbol{C}_i$ is not known, we usually have some assumptions about the relationship of prediction and localization labels. In this work, $Y_i^k = \mathcal{I}\left(\sum_{1 \leq j \leq T} C_i^{jk} \geq \eta\right)$, where $\mathcal{I}$ is an indicator function and $\eta$ is a constant which depends on the MIL assumption. For example, in our concept annotation tasks we make the standard assumption, i.e we assume $\eta = 1$, i.e. the time series label (bag label) for a concept is positive if that concept is present at any one time-stamp (at-least one instance has positive label).

### 3.3 RELATIONAL MULTI-INSTANCE LEARNING FRAMEWORK

Inspired by the recent success of recurrent neural networks in sequence modeling Bahdanau et al. (2014); Sutskever et al. (2014) and classification tasks Krizhevsky et al. (2012); Simonyan & Zisserman (2014), we adapt these models to the MIL framework to model multivariate time series data for concept annotation tasks. We denote all the variables at every time step as an instance and the entire multivariate time series as a bag. Unlike the traditional MIL setting, where the instances within a bag are independent of each other, in our case, the instances have relationships (namely temporal dependencies) among them. To model these dependencies, we propose to combine RNN models such as Long-Short Term Memory Neural Networks (LSTM) and Sequence-to-Sequence models with MIL, and propose our **R**elational **M**ulti-**I**nstance **L**earning framework, abbreviated as **RMIL**. RMIL takes in multivariate time series as input and outputs concept annotations. In RMIL, the outputs of RNN model provide the instance label predictions (i.e. solution for concept localization task) and the aggregation of the instance labels using aggregators such as pooling layer provides the bag label predictions (i.e. solution for concept prediction task). We propose different pooling functions and attention mechanisms which can be easily incorporated into our RMIL to improve the concept annotations.

**Pooling Layers for RMIL**  The bag-level prediction is obtained by using an aggregation gathering on all instance-level predictions. The aggregator function $g(\cdot) : [0,1]^T \mapsto [0,1]$ in RMIL can be modeled using the pooling layers. Without loss of generality we assume that RNN model computes a mapping from the feature time series to the concept time series for each of the concept $k \in \mathcal{C}$. Let us denote the probability of an instance $j$ belonging to concept $k$ as $p_{jk}$. Then, the bag level probability for a concept $k$ is given by $P_k = g(p_{1k}, p_{2k}, \ldots, p_{Tk})$. The role of aggregator function

$g(\cdot)$ is to combine the instance probabilities from each class specific feature map $\{p_{jk}\}$ into a single bag probability $P_k$. Several pooling mechanisms shown in Table 1 have been introduced in MIL and deep learning literature which can be used in our RMIL. In Table 1, $r$, $a$, $b_k$, and $r_k$ are parameters which can be fixed or are learned during training, and $\sigma(\cdot)$ denotes the sigmoid function.

**Table 1:** Pooling functions for RMIL.

| **Pooling Functions** | |
| --- | --- |
| Noisy-OR pooling (Zhang et al., 2006) | $P_k = g\left(\{p_{jk}\}\right) = 1 - \prod_j (1 - p_{jk})$ |
| Log-sum-exponention (LSE) (Ramon & De Raedt, 2000) | $P_k = g\left(\{p_{jk}\}\right) = \frac{1}{r} \log\left(\frac{1}{T} \sum_j \exp\left(r p_{jk}\right)\right)$ |
| Generalized Mean (GM) (Keeler et al., 1991) | $P_k = g\left(\{p_{jk}\}\right) = \left(\frac{1}{T} \sum_j p_{jk}^r\right)^{1/r}$ |
| Integrated segmentation and recognition (ISR) (Keeler et al., 1991) | $P_k = g\left(\{p_{jk}\}\right) = \left(\sum_j \frac{p_{jk}}{1-p_{jk}}\right) \Big/ \left(1 + \sum_j \frac{p_{jk}}{1-p_{jk}}\right)$ |
| Noisy-AND pooling (Kraus et al., 2016) | $P_k = g\left(\{p_{jk}\}\right) = \frac{\sigma\left(a\left(\sum_j p_{jk}/T - b_k\right)\right) - \sigma(-ab_k)}{\sigma(a(1-b_k)) - \sigma(-ab_k)}$ |
| Max pooling | $P_k = g\left(\{p_{jk}\}\right) = \max_j\left(p_{jk}\right)$ |
| Softmax pooling | $P_k = g\left(\{p_{jk}\}\right) = \left(\sum_j p_{jk} \exp\left(r_k p_{jk}\right)\right) \Big/ \left(\sum_j \exp\left(r_k p_{jk}\right)\right)$ |

**Attention Mechanism for RMIL**  Instances within each bag have temporal relations between them. We can use attention mechanism to focus on some of the instances and their relations to improve their instance-level predictions. In order to make predictions at time $j$, the hidden state $\boldsymbol{h}_j \in \mathbb{R}^Q$ of RNN can be used, where $Q$ is the hidden state dimension. However, relevant information may be captured by hidden states at other time steps as well. Thus, we may want to introduce an attention vector or matrix $(\boldsymbol{a})$ to leverage information of hidden states $\boldsymbol{H} = (\boldsymbol{h}_1, \cdots, \boldsymbol{h}_T)^\top \in \mathbb{R}^{T \times Q}$ from all time steps. Let us denote the output after attention as $\tilde{\boldsymbol{H}} \in \mathbb{R}^{T \times Q}$. The attention matrix can then be modeled using $\tilde{\boldsymbol{H}}$ in various ways as listed below.

*Feature-based Attention*  One idea is to design the attention matrix based on the feature and its time-stamp. Let us define a feature-based attention matrix as $\boldsymbol{A} = (\boldsymbol{a}_1, \cdots, \boldsymbol{a}_T)^\top \in \mathbb{R}^{T \times Q}$. For each $j = 1, \cdots, T$, we have

$$\boldsymbol{a}_j = \left(\exp\left(\boldsymbol{w}_j^\top \boldsymbol{H}\right)\right) \oslash \left(\sum_{1 \leq j' \leq T} \exp\left(\boldsymbol{w}_{j'}^\top \boldsymbol{H}\right)\right)$$

and $\tilde{\boldsymbol{H}} = \boldsymbol{A} \odot \boldsymbol{H}$, where $\odot$ and $\oslash$ are element-wise multiplication and division, respectively and $\boldsymbol{W} = (\boldsymbol{w}_1, \cdots, \boldsymbol{w}_T)^\top \in \mathbb{R}^{T \times T}$ is the weight matrix which can be learned during training. We call this *Attention-F* mechanism. We can simplify this attention by averaging the attentions for all hidden dimensions by taking $a_{jq} \leftarrow 1/Q \cdot \sum_{1 \leq q' \leq Q} a_{jq'}$. We will denote this as *Attention-FS* mechanism.

*Time-based Attention*  Attention model (Ma et al., 2017) can be designed to capture the relation between the current time step $j$ and previous time steps $j' \leq j$, by solely relying on previous hidden states $\boldsymbol{h}_{j'}$. We can define a time-based attention matrix as $\boldsymbol{A} \in \mathbb{R}^{T \times T}$. For each $j$ and $j'$ in $[1, \cdots, T]$, we have

$$a_{j,j'} = \begin{cases} \left(\exp\left(\boldsymbol{w}^\top \boldsymbol{h}_{j'}\right)\right) \Big/ \left(\sum_{1 \leq j'' \leq j} \exp\left(\boldsymbol{w}^\top \boldsymbol{h}_{j''}\right)\right), & j' \leq j; \\ 0, & \text{otherwise.} \end{cases}$$

and $\tilde{\boldsymbol{H}} = \boldsymbol{A} \cdot \boldsymbol{H}$, where $\boldsymbol{w} \in \mathbb{R}^D$ is the weight vector to learn. We use *Attention-T* to represent Time-based attention mechanism.

*Interaction-based Attention*    The time-based attention can be further improved by considering both the previous and current hidden states $\boldsymbol{h}_{j'}$ and $\boldsymbol{h}_j$ (Ma et al., 2017). In this case, we have

$$
a_{j,j'} = \begin{cases} \left(\exp\left(\boldsymbol{v}^\top \tanh(\boldsymbol{W_1}\boldsymbol{h}_j + \boldsymbol{W_2}\boldsymbol{h}_{j'})\right)\right) / \left(\sum_{1 \leq j'' \leq j} \exp\left(\boldsymbol{v}^\top \tanh(\boldsymbol{W_1}\boldsymbol{h}_j + \boldsymbol{W_2}\boldsymbol{h}_{j''})\right)\right), & j' \leq j; \\ 0, & \text{otherwise} \end{cases}
$$

and similarly $\tilde{\boldsymbol{H}} = \boldsymbol{A} \cdot \boldsymbol{H}$. Here, we need to learn $\boldsymbol{v} \in \mathbb{R}^S, \boldsymbol{W_1} \in \mathbb{R}^{S \times Q}, \boldsymbol{W_2} \in \mathbb{R}^{S \times Q}$, and we choose $S = Q/2$. A simplified version of interaction-based attention can be obtained if we use vector $\boldsymbol{w_1} \in \mathbb{R}^Q, \boldsymbol{w_2} \in \mathbb{R}^Q$ instead of matrices $\boldsymbol{W_1}, \boldsymbol{W_2}$ and by setting $\boldsymbol{v} = \boldsymbol{1}$ in the above equation. We use *Attention-I* and *Attention-IS* to represent Interaction-based attention mechanism and simplified version of interaction-based attention mechanism respectively.

The above attention mechanisms usually help both prediction and localization tasks.

## 4 Experiments

Here, we demonstrate the performance of our proposed RMIL models on concept annotation tasks i.e. concept prediction and localization tasks, using a real-world health-care dataset and compare its performance to the popular multi-instance learning approaches. In addition, we discuss the impact of using pooling functions and attention mechanism in our RMIL framework.

### 4.1 Dataset Descriptions and Experimental Design

To evaluate our RMIL, we ran experiments on MIMIC-III RESP datasets whose statistics is shown in Table 2.

**Table 2:** MIMIC-III RESP dataset.

|  | **MIMIC-III RESP** |
| --- | --- |
| # of samples ($N$) | 2014 |
| # of variables ($D$) | 21 |
| # of time steps | 4 |
| # of concepts | 26 |

**MIMIC-III RESP Dataset**    MIMIC-III is a public dataset (Johnson et al., 2016) which has deidentified clinical care data collected at Beth Israel Deaconess Medical Center from 2001 to 2012. It contains over 58,000 hospital admission records of 38,645 adults and 7,875 neonates. For our work, we extracted 21 feature time series from more than 2,000 adult patients who were diagnosed with a respiratory disorder such as Acute Hypoxemic Respiratory Failure (AHRF) (Khemani et al., 2009) at the time of admission. These 21 features are respiratory based features such as peak inspiratory pressure (PIP) and arterial partial pressure of oxygen (PaO2) and were collected during the first 3 days after admission. The feature time series has 4 time stamps and the first time stamp corresponds to the admission time. We denote this dataset as MIMIC-III RESP dataset. In addition, we also generated another feature time series with more time stamps whose results is shown in the appendix.

**Concept Annotations**    The medical time series data of MIMIC-III dataset does not come with the concept annotations, however the medical concepts are available in the doctor notes of the MIMIC-III database. To obtain the concept annotations, we extract the concept time series from the doctor notes using the `NOTEEVENTS` table of MIMIC-III database. The total number of doctor notes is 2,083,180, out of which 98.15% of notes (2,044,634 notes) have no timestamp and 1.85% of notes (38,546 notes) have timestamps associated with them. The total number of unique concepts in the doctor notes in the first 3 days data is 6,197. To obtain concept time series for each patient with respiratory disorder such as AHRF, we first identified respiratory-related concepts from the medical literature (Antonelli et al., 2001; Khemani et al., 2009), and obtained their medical codes from the Unified Medical Language System (UMLS) dictionary (Bodenreider, 2004). Then, we mined the patient's doctor notes from `NOTEEVENTS` table to extract all the possible medical concepts related to the respiratory system and its disorder. In total, we chose top 26 respiratory concepts to generate concept time series which has the same number of time stamps as feature time series.

**Table 3:** Concept annotation results on MIMIC-III RESP dataset. Max-pooling function was used in all RMIL models.

| | | Prediction | | Localization | |
|---|---|---|---|---|---|
| | | AUROC | AUPRC | AUROC | AUPRC |
| RMIL Models | S2S | 0.858 | 0.756 | 0.788 | 0.431 |
| | S2S with Attention I | 0.858 | 0.758 | 0.793 | 0.433 |
| | LSTM | 0.860 | 0.763 | 0.795 | 0.444 |
| | LSTM with Attention I | 0.862 | 0.766 | 0.797 | 0.444 |
| | Bi-LSTM | 0.864 | 0.769 | 0.789 | 0.415 |
| | Bi-LSTM with Attention I | 0.864 | 0.769 | 0.796 | 0.420 |
| MIL Models | CNN | 0.857 | 0.756 | 0.785 | 0.397 |
| | CNN with Attention I | 0.855 | 0.755 | 0.785 | 0.409 |
| | DPMIL | 0.531 | 0.222 | 0.516 | 0.148 |
| | MISVM | 0.751 | 0.613 | 0.706 | 0.333 |

## 4.2 COMPARISON AND IMPLEMENTATION DETAILS

We compare the performance of our proposed models to the popular MIL models such as MISVM (Andrews et al., 2003), DPMIL (Kandemir & Hamprecht, 2014) and Convolutional Neural Networks (CNN) Kraus et al. (2016), and CNN with attention. We categorize all the evaluated methods into two groups:

1. *Multi-Instance Learning models (MIL)*: We treat MISVM, DPMIL, Convolutional Neural Networks (CNN), and CNN with Attention I as our baseline MIL models.
2. *Relational Multi-Instance Learning models (RMIL)*: We evaluate the following deep learning models as part of our RMIL framework:
   (a) Long Short-Term Memory neural networks (LSTM) (Hochreiter & Schmidhuber, 1997)
   (b) Bi-directional LSTM (Bi-LSTM) (Graves et al., 2013)
   (c) Sequence to Sequence models (S2S) (Sutskever et al., 2014)
   (d) The above three RMIL models with different attention mechanisms
   (e) The above RMIL models with different pooling functions

For LSTM models, we use two LSTM layers and two dense layers. For S2S models, we use two LSTM layers for both the encoder and the decoder. For Bi-LSTM models, we use two bi-directional LSTM layers. All the models were constructed to have a comparable number of parameters. We train all the Deep learning models with the RMSProp optimization method and we use early stopping to find the best weights on the validation dataset. For baseline MIL models, we follow the suggestions of the corresponding papers to fine-tune the parameters. All the input variables in the training data are normalized to be 0 mean and 1 standard deviation. The inputs to all the models is the same feature time series data. We used Keras (Chollet, 2017) and Python to run the deep models and MISVM models. Matlab code from the original authors was used to obtain DPMIL results. We use the area under ROC (AUROC) and area under precision-recall curve (AUPRC) scores as our evaluation metrics and report the results from 5-fold cross validation for all the evaluated methods.

## 4.3 QUANTITATIVE RESULTS

Table 3 shows the concept annotation results on the MIMIC-III RESP dataset. From this table, we see that RMIL models outperform the non-deep multi-instance learning models by at least 8-10% for concept localization task, and by at least 10-15% for concept prediction tasks in terms of AUROC and AUPRC. RMIL performs slightly better than CNN-based models on all the metrics. Among all the RMIL models, we find that LSTM model obtains slightly better overall results compared to the other models for localization task.

To study the impact of pooling and attention, we trained and evaluated LSTM models with different pooling functions and different attention mechanisms, which are described in Section 3. Tables 4 and 5 show the comparison results. From these tables, we observed that (i) all the attention mechanisms except feature-based attention perform similar to each other especially for the prediction task, and (ii) all the pooling functions other than ISR and Noisy-OR obtain similar overall performance. This

**Table 4:** Results on RMIL LSTM models with different pooling functions and with Attention-I mechanism.

| | Prediction | | Localization | |
|---|---|---|---|---|
| | **AUROC** | **AUPRC** | **AUROC** | **AUPRC** |
| LSTM with ISR Pooling | 0.862 | 0.765 | 0.733 | 0.375 |
| LSTM with Noisy-AND Pooling | 0.863 | 0.767 | 0.779 | 0.431 |
| LSTM with Generalized Mean Pooling | 0.863 | 0.767 | 0.796 | 0.450 |
| LSTM with LSE Pooling | 0.863 | 0.767 | 0.792 | 0.452 |
| LSTM with Softmax Pooling | 0.863 | 0.767 | 0.790 | 0.450 |
| LSTM with Noisy-OR Pooling | 0.860 | 0.762 | 0.703 | 0.318 |
| LSTM with Max Pooling | 0.862 | 0.766 | 0.797 | 0.444 |

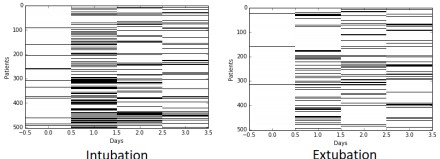 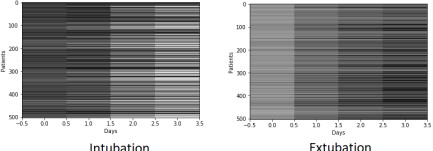

(a) Ground truth annotations of intubation and extubation concepts.

(b) Intubation and extubation concept prediction probabilities identified by attention-based LSTM.

**Figure 2:** Ground truth and predicted concept annotation comparison. In 2(a) white block corresponds to absence of a concept and black block corresponds to presence of a concept. In 2(b), darker gray value indicates higher chance for presence of a concept. X-axis represents time steps, Y-axis corresponds to different patients.

demonstrates that choice of attention does not matter but choice of pooling has some impact in our RMIL framework.

**Table 5:** Results on RMIL LSTM models with different attention mechanisms and with max pooling.

| | Prediction | | Localization | |
|---|---|---|---|---|
| **Model** | **AUROC** | **AUPRC** | **AUROC** | **AUPRC** |
| LSTM with Attention-T | 0.864 | 0.768 | 0.796 | 0.445 |
| LSTM with Attention-IS | 0.862 | 0.767 | 0.798 | 0.446 |
| LSTM with Attention-I | 0.862 | 0.766 | 0.797 | 0.444 |
| LSTM with Attention-FS | 0.861 | 0.765 | 0.796 | 0.417 |
| LSTM with Attention-F | 0.861 | 0.764 | 0.780 | 0.408 |
| LSTM with no attention | 0.860 | 0.763 | 0.795 | 0.444 |

### 4.4 DISCUSSIONS

We can study the interpretability of concept localization by looking at the localization results of our RMIL models, even though the model is trained without the labels for localization. Figure 2 shows the ground truth annotations of two respiratory concepts - intubation and extubation concepts, and the prediction probabilities of these concepts obtained by our RMIL attention-based LSTM models. From figure 2(a) we can make the following observations, (i) intubation usually happens before extubation for the same patient, (ii) intubation and extubation could happen on the same day, and (iii) intubation and extubation occur commonly within the first 24 hours of admission. From the figure 2(b), we see that our RMIL attention based LSTM predicts that the probability of intubation happening on the first day of admission is higher (draker gray means higher probability of concept occurrence) and the probability of extubation happening within first day is lower. This indicates that the model has correctly learnt that intubation should appear before extubation. This also implicitly implies that the RMIL attention-based LSTM models have correctly learnt the instance-level relationships from the medical time series data with only bag-level labels.

## 5 SUMMARY

In this paper, we presented Relational Multi-Instance Learning - a deep multi-instance learning framework using recurrent neural networks for concept annotation from the medical time series data. Empirical results on medical dataset demonstrated that our proposed models outperform the popular state-of-the-art multi-instance learning approaches. Experiments with different pooling and attention mechanisms showed that while attention mechanism does not have a significant impact on model's performance, certain pooling functions such as ISR and Noisy-OR can negatively impact the instance prediction results.

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

# 6 APPENDIX

## 6.1 MIMIC-III CONCEPT LIST

Table 6 show the concept list of the MIMIC-III RESP dataset used in our experiments.

## 6.2 ANOMALY DETECTION FOR TIME SERIES

Here, we will demonstrate anomaly detection from medical time series data as a use case application of our RMIL framework.

### 6.2.1 MOTIVATION AND RELATED WORK

Each year, more than 1,000,000 US adults and children are put on mechanical ventilation during their stays in ICU. However, lack of effective tools to aid with ventilator weaning and extubation (removal of the breathing tube) readiness assessment results in nearly half of the patients spending unnecessary days on ventilators (Randolph et al., 2002), and up to 20% of them having ventilators discontinued too soon (Kurachek et al., 2003). Spending unnecessary days on ventilators can lead to hospital-acquired infections while having ventilators discontinued too soon could require painful reintubation. A work-of-breathing measure called Pressure-Rate Product (PRP), calculated from esophageal pressure, has shown potential to be used as a guideline for ventilator weaning and extubation readiness assessment (Willis et al., 2005). However, PRP calculations are susceptible to sensor artifacts and breathing pattern anomalies. These anomalies limit realtime use of PRP for clinical decision making. Our goal is to automatically detect and remove these anomalies by using our proposed Relational Multi-instance learning models, thereby enabling real-time clinical decision making. Figure 3 shows the example of anomaly detection from the ventilator time series data. Here, anomalies appear due to patient factors (cough, movement) and instrument factors (probing, catherter drift), and should be automatically detected from the ventalitor monitoring time series data.

**Table 6:** MIMIC-III respiratory concept list.

| Concept Number | Concept Name |
|---|---|
| 1 | Respiratory rate |
| 2 | Biomedical tube device |
| 3 | Analysis of arterial blood gases and ph |
| 4 | Ventilation, function (observable entity) |
| 5 | Medical history |
| 6 | Injury wounds |
| 7 | Body weights |
| 8 | Tidal volume |
| 9 | Inspired fraction of oxygen |
| 10 | Intubation |
| 11 | Positive end-expiratory pressure |
| 12 | Artificial airways |
| 13 | Bicarbonates |
| 14 | Tracheal extubation |
| 15 | Chest radiograph |
| 16 | Mean blood pressure |
| 17 | Oxygen measurement, partial pressure, arterial |
| 18 | Cardiac arrest |
| 19 | Peak inspiratory pressure |
| 20 | Biomechanical compliance |
| 21 | Tachycardia ventricular |
| 22 | Scoliosis |
| 23 | Acute lung injury |
| 24 | Partial pressure of carbon dioxide in arterial blood |
| 25 | Resuscitate |
| 26 | Sudden onset |

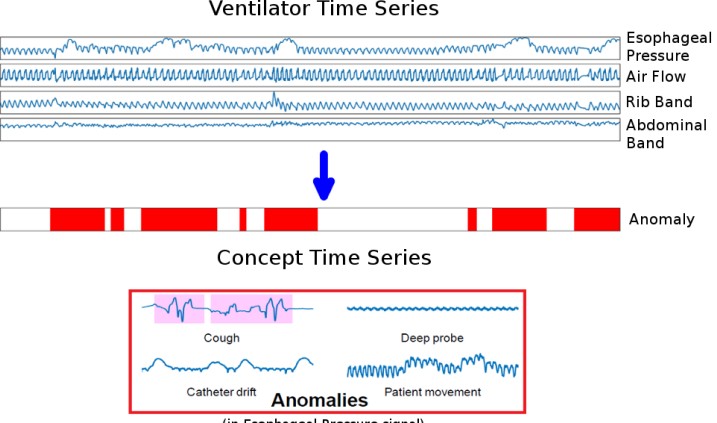

**Figure 3:** Anomaly detection from ventilator time series data.

There is a long history and a rich body of research work on anomaly detection in time series data. See Hodge & Austin (2004); Chandola et al. (2009) for a quick survey on generic anomaly detection algorithms. Gupta et al. (2014) also author a survey on anomaly detection for time series data. In their follow-up survey, Chandola et al. (2012) gave a summary on discrete sequence anomaly detection algorithms. Unfortunately, techniques for time series anomaly detection are quite domain-specific. This is due to highly-varied nature of time series characteristics. In Chan & Mahoney (2005), a set of $k$ minimum bounding rectangles between each time step is used as a model of normal data generating distribution. In Camerra et al. (2010); Lin et al. (2003), continuous time series are converted to ordinal symbolic representation, allowing faster approximation of Euclidean distance between time series windows and early pruning. Then, anomaly detection can be performed by setting a threshold

on distance value. Recently, Wang et al. (2016) proposes a self-learning method that learns clusters of constrained grammars based on ordinal symbolic approximation of time series value. Guigou et al. (2017) proposes incorporating experts into anomaly detection in a method inspired by immune system. In Jones et al. (2016), the authors propose extracting exemplars from Euclidean pairwise distance as a way to speed up anomaly detection algorithm. Most of these techniques for time series anomaly detection are quite domain-specific, and few of them model the anomaly detection problem in Multi-instance learning setting.

### 6.2.2 PROBLEM FORMULATION

We formulate the anomaly detection as concept annotation problem in MIL setting, where the bag corresponds to the medical time series data $X_i$ and an instance is the features at a time-stamp, and the anomaly corresponds to a concept. Thus, the prediction (predicting $Y_i$) and localizaton tasks (predicting $C_i$) correspond to predicting the presence and location of the anomalies in the time series data. For anomaly detection, we assume $Y_i^k = \mathcal{I}\left(\sum_{1 \leq j \leq T} C_i^{jk} \geq \eta\right)$, where $\eta$ is chosen as $\eta = 0.6T$ and $T$ is the number of time series windows within the bag.

### 6.2.3 EXPERIMENTS

We conduct anomaly detection experiments on a PICU dataset described below.

**PICU Dataset** This dataset consists of 1,530 recorded 5-minute 200-Hz sensor readings from 385 subjects collected at a leading children's hospital by a team of research clinicians. Statistics of the dataset is shown in Table 7. The recordings are made on mechanically-ventilated patients in pediatrics ICU ward. Medical time series data collected from four sensors are used: flow volume spirometry, esophageal pressure sensor, and dual band respiratory inductance plethysmography. Each subject can be under one of four breathing conditions: ventilated

**Table 7:** PICU dataset statistics.

|  | **PICU** |
| --- | --- |
| # of samples ($N$) | 1530 |
| # of variables ($D$) | 80 |
| # of time steps | 60 |
| # of concepts | 1 |

with Continuous Positive Airway Pressure (CPAP), ventilated with Pressure Support (PS), 5 minutes after extubation, and 60 minutes after extubation. Along with the 4 sensor signals, clinicians verified binary anomaly label generated using hard-crafted state-of-the-art breathing anomaly detection algorithm is provided as ground truth.

We annotate the concept of breathing anomaly in MIL framework by processing the dataset as follows. The sensor recordings are split into non-overlapping 5-second windows. In MIL framework, each window becomes an instance and each recording becomes a bag. For each sensor signal of each window, we extract 20 Mel-Frequency Cepstral Coefficients (MFCCs) to be used as features of each instance; thus, each instance has 80 features in total. For each instance, its anomaly label is set to positive if at least 20% of the window are labeled as anomaly. The bag anomaly label is set to positive if at least 60% of its instances are labeled as anomaly.

**Table 8:** Anomaly detection results on PICU dataset with different attention mechanisms and with max pooling.

|  |  | **Prediction** | | **Localization** | |
| --- | --- | --- | --- | --- | --- |
|  |  | **AUROC** | **AUPRC** | **AUROC** | **AUPRC** |
| | S2S | 0.741 | 0.797 | 0.532 | 0.707 |
| | S2S with Attention-I | 0.732 | 0.787 | 0.511 | 0.694 |
| | S2S with Attention-F | 0.730 | 0.789 | 0.519 | 0.689 |
| | LSTM | 0.708 | 0.763 | 0.543 | 0.712 |
| RMIL Models | LSTM with Attention-I | 0.690 | 0.767 | 0.513 | 0.682 |
| | LSTM with Attention-F | 0.712 | 0.776 | 0.522 | 0.690 |
| | Bi-LSTM | 0.729 | 0.784 | 0.562 | 0.725 |
| | Bi-LSTM with Attention-I | 0.711 | 0.769 | 0.535 | 0.708 |
| | Bi-LSTM with Attention-F | 0.736 | 0.790 | 0.539 | 0.704 |
| MIL Models | Cluster-MIL* | 0.590 | 0.620 | 0.530 | 0.670 |
| | MISVM | 0.572 | 0.647 | 0.542 | 0.606 |

**Results**    Table 8 shows the anomaly detection results using our RMIL models. We observe that (i) RMIL models mostly outperform the non-deep MIL models on both tasks, (ii) LSTM and Bi-LSTM based RMIL models obtain better overall results compared to the sequence-to-sequence models, and (iii) the attention mechanism does not help for localization task but sometimes obtains better results for the prediction task.

*Remark:* Cluster-MIL* is a HDBSCAN clustering (Campello et al., 2015) based approach for anomaly detection. It uses both instance and bag labels for training, while other models only use bag labels for training.

## 6.3    ADDITIONAL RESULTS ON MIMIC-III RESP DATASET

We sampled the feature time series from MIMIC-III RESP dataset every 6 hours and generated a time series with more (12) time stamps. We call this MIMIC-III RESP-II dataset.

Table 9 shows the results of RMIL models on this dataset. We observe that (i) All the RMIL models have similar performance for prediction task, (ii) Bi-LSTM RMIL has better localization results compared to other the models.

**Table 9:** MIMIC-III RESP-II dataset with different attention mechanisms and with max pooling.

|  |  | Prediction | | Localization | |
| --- | --- | --- | --- | --- | --- |
|  |  | **AUROC** | **AUPRC** | **AUROC** | **AUPRC** |
|  | S2S | 0.874 | 0.746 | 0.787 | 0.185 |
|  | S2S with Attention-I | 0.875 | 0.749 | 0.794 | 0.186 |
|  | S2S with Attention-F | 0.869 | 0.739 | 0.777 | 0.188 |
|  | LSTM | 0.875 | 0.749 | 0.776 | 0.177 |
| RMIL Models | LSTM with Attention-I | 0.875 | 0.750 | 0.781 | 0.175 |
|  | LSTM with Attention-F | 0.872 | 0.742 | 0.771 | 0.177 |
|  | Bi-LSTM | 0.876 | 0.749 | 0.801 | 0.206 |
|  | Bi-LSTM with Attention-I | 0.875 | 0.750 | 0.803 | 0.204 |
|  | Bi-LSTM with Attention-F | 0.871 | 0.741 | 0.769 | 0.171 |

