# OpenReview forum: "Relational Multi-Instance Learning for Concept Annotation from Medical Time Series"
_ICLR.cc/2018/Conference — Reject_

### Official Review · AnonReviewer2 · 2017-11-20
**Overall, this is a reasonable paper with no obvious major flaws.  The novelty and impact may be greater on the application side than on the methodology side.**

**Rating:** 6
**Confidence:** 4

**Review:**

The paper addresses the classification of medical time-series data by formulating the problem as a multi-instance learning (MIL) task, where there is an instance for each timestep of each time series, labels are observed at the time-series level (i.e. for each bag), and the goal is to perform instance-level and series-level (i.e. bag-level) prediction.  The main difference from the typical MIL setup is that there is a temporal relationship between the instances in each bag.  The authors propose to model this using a recurrent neural network architecture.  The aggregation function which maps instance-level labels to bag-level labels is modeled using a pooling layer (this is actually a nice way to describe multi-instance classification assumptions using neural network terminology).  An attention mechanism is also used.

The proposed time-series MIL problem formulation makes sense.  The RNN approach is novel to this setting, if somewhat incremental.  One very positive aspect is that results are reported exploring the impact of the choice of recurrent neural network architecture, pooling function, and attention mechanism.  Results on a second dataset are reported in the appendix, which greatly increases confidence in the generalizability of the experiments.  One or more additional datasets would have helped further solidify the results, although I appreciate that medical datasets are not always easy to obtain.  Overall, this is a reasonable paper with no obvious major flaws.  The novelty and impact may be greater on the application side than on the methodology side.

Minor suggestions:

-The term "relational multi-instance learning" seems to suggest a greater level of generality than the work actually accomplishes.  The proposed methods can only handle time-series / longitudinal dependencies, not arbitrary relational structure.  Moreover, multi-instance learning is typically viewed as an intermediary level of structure "in between" propositional learning (i.e. the standard supervised learning setting) and fully relational learning, so the "relational multi-instance learning" terminology sounds a little strange. Cf.:
De Raedt, L. (2008). Logical and relational learning. Springer Science & Business Media.

-Pg 3, a capitalization typo: "the Multi-instance learning framework"

-The equation for the bag classifier on page 4 refers to the threshold-based MI assumption, which should be attributed to the following paper:
Weidmann, N., Frank, E. & Pfahringer, B. 2003. A two-level learning method for generalized multi-instance problems. In Proceedings of the 14th European Conference on Machine Learning,
Springer, 468-479.
(See also: J. R. Foulds and E. Frank. A review of multi-instance learning assumptions. Knowledge Engineering Review, 25(1):1-25, 2010. )

- Pg 5, "Table 1" vs "table 1" - be consistent.

-A comparison to other deep learning MIL methods, i.e. those that do not exploit the time-series nature of the problem, would be valuable.  I wouldn't be surprised if other reviewers insist on this.

---

> ### Author Response · Authors · 2018-01-03
> **Re: Overall, this is a reasonable paper with no obvious major flaws. The novelty and impact may be greater on the application side than on the methodology side.**
>
> >> Many Thanks for your encouraging feedback. We have fixed the typos, cited the suggested references and incorporated your suggestions in our revised draft.
>
> - Comparison with other deep learning baselines: We have tried other deep learning baselines such as CNN, and CNN with attention; however, their performance was slightly worse than the RNN models. CNN model obtained AUROC and AUPRC of [0.857,0.756] and [0.785,0.397] for concept prediction and localization tasks respectively. While, CNN with attention model obtained AUROC and AUPRC of about [0.855, 0.755] and [0.785, 0.409] for concept prediction and localization tasks respectively. We have included these results in the revised draft.

---

### Official Review · AnonReviewer3 · 2017-11-27
**Relational Multi-Instance Learning for Concept Annotation from Medical Time Series**

**Rating:** 3
**Confidence:** 3

**Review:**

This paper proposes a framework called 'multi-instance learning', in which a time series is treated as a 'set' of observations, and label is assigned to the full set, rather than individual observations. In this framework, authors propose to do set-level prediction (using pooling) and observation level predictions (using various attention mechanisms).
They test their approach in a medical setting, where the goal is to annotate vital signs time series by clinical events. Their cohort is 2014 adults time series (average length 4 time steps), and their time series has dimension of 21 and their clinical events have dimension of 26. Their baselines are other 'multi-instance learning' prior work and results are achieved through cross-validation. A few of the relevant hyper-parameters are tuned and some important hyper-parameters (i.e. number of hidden states in the LSTMs, or optimization method and learning rate) are not tuned.

Originality - I find the paper to be very incremental in terms of originality of the method.

Quality and Significance - Due to small size of the cohort and lack of additional dataset, it is difficult to reliably access quality of experiments. Given that results are reported via cross-validation and without a true held-out dataset, and given that a number of hyperparameters are not even tuned, it is difficult to be confident that the differences of all the methods reported are significant.

Clarity - The writing has good clarity.

Major issues with the paper:
- Lack of reliable experiment section. Dataset is too small (2000 total samples), and model training is not described with enough details in terms of hyper-parameters tuned.

---

> ### Author Response · Authors · 2018-01-03
> **Re: Relational Multi-Instance Learning for Concept Annotation from Medical Time Series**
>
> Thank you for your comments and suggestions. Please find our response below:
>
> Originality - I find the paper to be very incremental in terms of originality of the method.
> >> We agree that the ideas presented here are simple. However, we want to point out that the simple way of looking at RNNs in MIL settings has not been presented before to the best of our knowledge. That is, there is no existing RNN work which is trained with overall labels but aims for labels at each time steps.  Also, we show that such a frustratingly simple RNN model can achieve excellent performance compared to the other existing MIL approaches.
>
> Quality and Significance - Due to small size of the cohort and lack of additional dataset, it is difficult to reliably access quality of experiments. Given that results are reported via cross-validation and without a true held-out dataset, and given that a number of hyperparameters are not even tuned, it is difficult to be confident that the differences of all the methods reported are significant.
> >> We have conducted exhaustive experiments to fine-tune the model’s hyper-parameters, and found that all the models achieve similar performance as reported in our paper.
>
> Major issues with the paper:
> - Lack of reliable experiment section. Dataset is too small (2000 total samples), and model training is not described with enough details in terms of hyper-parameters tuned.
> >> This was the biggest dataset we could obtain under our problem settings by mining one of the largest publicly available healthcare dataset called MIMIC-III [1]. Thus, we believe the dataset size is reasonable given the data source and application domain. Also, we have provided additional results on a different dataset for the anomaly detection problem using the RMIL framework in the appendix of our paper. Kindly note that unlike other application domains, in medical domain the dataset sizes are relatively smaller.
>
> [1] AEW Johnson, TJ Pollard, L Shen, L Lehman, M Feng, M Ghassemi, B Moody, P Szolovits, LA Celi,
> and RG Mark. Mimic-iii, a freely accessible critical care database. Scientific Data, 2016.

---

### Official Review · AnonReviewer1 · 2017-11-29
**A new MIL formulation with limited technical innovation and unconvincing experimental results.**

**Rating:** 3
**Confidence:** 5

**Review:**

==== Post Rebuttal ====
I went through the rebuttal, which unfortunately claimed a number statements without any experimental support as requested. The revision didn't address my concerns, and I've lowered my rating.

==== Original Review ====
This paper proposed a novel Multiple Instance Learning (MIL) formulation called Relation MIL (RMIL), and discussed a number of its variants with LSTM, Bi-LSTM, S2S, etc. The paper also explored integrating RMIL with various attention mechanisms, and demonstrated its usage on medical concept prediction from time series data.

The biggest technical innovation in this paper is it combines recurrent networks like Bi-LSTM with MIL to model the relations among instances. Other than that, the paper has limited technical innovations: the pooling functions were proposed earlier and their integration with MIL was widely studied before (as cited by the authors); the attention mechanisms are also proposed by others.

However, I am doubtful whether it’s appropriate to use LSTM to model the relations among instances. In general MIL, there exists no temporal order among instances, so modeling them with a LSTM is unjustified. It might be acceptable is the authors are focusing on time-series data; but in this case, it’s unclear why the authors are applying MIL on it. It seems other learning paradigm could be more appropriate.

The biggest concern I have with this paper is the unconvincing experiments. First, the baselines are very weak. Both MISVM and DPMIL are MIL methods without using deep learning features. It them becomes very unclear how much of the gain on Table 3 is from the use of deep learning, and how much is from the proposed RMIL.

Also, although the authors conducted a number of ablation studies, they don’t really tell us much. Basically, all variants of the algorithm perform as well, so it’s confusing why we need so many of them, or whether they can be integrated as a better model.

This could also be due to the small dataset. As the authors are proposing a new MIL learning paradigm, I feel they should experiment on a number of MIL tasks, not limited to analyzing time series medical data. The current experiments are quite narrow in terms of scope.

---

> ### Author Response · Authors · 2018-01-03
> **Re: A new MIL formulation with limited technical innovation and unconvincing experimental results.**
>
> Thank you for your useful comments.  Please find our response below:
>
> In general MIL, there exists no temporal order among instances, so modeling them with a LSTM is unjustified.
> >> Yes, in general MIL the temporal order is not modeled. However, in this paper, we are working with time series data which come with temporal dependencies, thus, we employ LSTM to model them in MIL setting which we refer to as Relational MIL.
>
> One of the key contributions of this work is to show that Recurrent Neural Network models such as  LSTM can be used in MIL setting with a suitable way to model instance-level and bag-level predictions. To the best of our knowledge, this simple way of looking at RNNs in MIL settings has not been presented before. Another point is to showcase that frustratingly simple RNN models can achieve excellent performance compared to the other MIL approaches.
>
> The biggest concern I have with this paper is the unconvincing experiments. First, the baselines are very weak. Both MISVM and DPMIL are MIL methods without using deep learning features.
> >> The baselines included in the paper are some of the popular and best performing baselines available for MIL framework. Both MISVM and DPMIL do not provide a way to model the relations between the instances as considered in the proposed RMIL. Thus, even if we use deep learning features with MISVM and DPMIL, they are bound to perform worse than the RMIL models since they do not model temporal dependencies present in the data. We will include these comparison results in our future work. In the revised draft we have included results from CNN models which obtain better results than MISVM and DPMIL,  but performs slightly worse than our RMIL models.
>
>
> Also, although the authors conducted a number of ablation studies, they don’t really tell us much. Basically, all variants of the algorithm perform as well, so it’s confusing why we need so many of them, or whether they can be integrated as a better model.
> >> As stated earlier we wanted to show that frustratingly simple RNN models can achieve excellent performance compared to the other MIL approaches. We have conducted exhaustive experiments on more complicated deep models, and have also tested a combination of several deep models; however, all our experiments showed that simple RNN models in RMIL framework achieve simpler results. In terms of an integration model, we’ve tried combinations / ensembles of different pooling layers / activation mechanisms, but we did not find any improvements in the performance.
>
> As the authors are proposing a new MIL learning paradigm, I feel they should experiment on a number of MIL tasks, not limited to analyzing time series medical data.
> >> Thanks for the suggestions. Our goal was to solve the time series prediction and localization problem as applicable to medical time series data. We have shown additional results of RMIL for anomaly detection task in the appendix. Unfortunately, conducting experiments outside of time series data is beyond the scope of this paper.

---

### Decision · Program_Chairs · 2018-01-29
**ICLR 2018 Conference Acceptance Decision**

**Decision:**

Reject

**Comment:**

This paper presents a MIL method for medical time series data. General consensus among reviewers that work does not meet criteria for being accepted.

Specifically:

Pros:
- A variety of meta-learning parameters are evaluated for the task at hand.
- Minor novelty of the proposed method

Cons:
- Minor novelty of the proposed method
- Rationale behind architectural design
- Thoroughness of experimentation
- Suboptimal choice of baseline methods
- Lack of broad evaluation across applications for new design
- Small dataset size
- Significance of improvement